# Vinorelbine Improves the Efficacy of Sorafenib against Hepatocellular Carcinoma: A Promising Therapeutic Approach

**DOI:** 10.3390/ijms25031563

**Published:** 2024-01-26

**Authors:** Wai Har Ng, Khee Chee Soo, Hung Huynh

**Affiliations:** 1Laboratory of Molecular Endocrinology, National Cancer Centre Singapore, Singapore 168583, Singapore; nmsnwh@nccs.com.sg; 2Division of Surgery and Surgical Oncology, National Cancer Centre Singapore, Singapore 168583, Singapore

**Keywords:** hepatocellular carcinoma, sorafenib, vinorelbine, antitumor growth, patient-derived xenograft, focal adhesion kinase

## Abstract

Hepatocellular carcinoma (HCC) is a leading global cause of cancer-related mortality. Despite the widespread adoption of sorafenib as the standard HCC treatment, its efficacy is constrained, frequently encountering resistance. To augment the effectiveness of sorafenib, this study investigated the synergy of sorafenib and vinorelbine using 22 HCC patient-derived xenograft (PDX) models. In this study, mice bearing HCC tumors were treated with the vehicle, sorafenib (15 mg/kg), vinorelbine (3 mg/kg), and sorafenib–vinorelbine combination (Sora/Vino). Rigorous monitoring of the tumor growth and side effects coupled with comprehensive histological and molecular analyses was conducted. The overall survival (OS) of mice bearing HCC orthotopic tumors was also assessed. Our data showed a notable 86.4% response rate to Sora/Vino, surpassing rates of 31.8% for sorafenib and 9.1% for vinorelbine monotherapies. Sora/Vino significantly inhibited tumor growth, prolonged OS of mice bearing HCC orthotopic tumors (*p* < 0.01), attenuated tumor cell proliferation and angiogenesis, and enhanced necrosis and apoptosis. The combination therapy effectively suppressed the focal adhesion kinase (FAK) pathway, which is a pivotal player in cell proliferation, tumor angiogenesis, survival, and metastasis. The noteworthy antitumor activity in 22 HCC PDX models positions Sora/Vino as a promising candidate for early-phase clinical trials, leveraging the established use of sorafenib and vinorelbine in HCC and other cancers.

## 1. Introduction

HCC is the second-most-common cancer worldwide, with an annual mortality rate of approximately 830,180 individuals [1]. While hepatic resection and liver transplantation offer potential curative options for early-stage HCC patients [2], more than 80% of HCC patients are initially diagnosed with advanced disease stages, and surgical interventions are not feasible [3]. Even in cases where surgery is possible, the 5-year survival rates remain relatively low due to the high likelihood of disease recurrence [4]. Given the aggressive nature of HCC and its propensity for early dissemination inside the liver [5], novel, effective, and affordable treatments for this lethal disease are urgently needed.

Sorafenib is a multikinase inhibitor that improves the median OS of patients with HCC [6] and has been approved as a standard treatment for this disease [6,7]. Despite its clinical benefits, its impact is modest and is often accompanied by the development of drug resistance. Therefore, it is crucial to explore new strategies to enhance the effectiveness of sorafenib and overcome drug resistance. As second-line treatments, the multikinase inhibitors regorafenib [8] and cabozantinib [9] were approved after showing significantly improved HCC patient survival compared to the placebo. Recently, lenvatinib has become a first-line alternative treatment for patients with unresectable HCC [10]. It is worth noting that systemic therapies with atezolizumab plus bevacizumab and durvalumab plus tremelimumab are considered as preferable to first-line therapy options [11], except for patients with high-risk stigmata of variceal or other gastrointestinal bleeding and those with liver cirrhosis in whom immune-based regimes are contraindicated (e.g., severe autoimmune disorders or liver transplantation) [12].

Vinorelbine, classified as a low-molecular-weight vascular disrupting agent [13] and a semi-synthetic alkaloid, is widely used to inhibit microtubule polymerization in cancer cells [14,15]. Previous studies have shown that administering vinorelbine in a metronomic manner results in objective responses of prolonged duration and minimal toxicity [16,17]. Although the safety and clinical efficacy demonstrated in trials have established vinorelbine as a standard concurrent chemo-radiotherapy regimen for various cancers [18,19], a phase I study indicated that patients with moderate and severe liver dysfunction experienced high-grade toxicities and poor tolerance when vinorelbine was administered at doses of 15 mg/m^2^ and ≥7.5 mg/m^2^, respectively [20]. Thus, it is recommended to evaluate the levels of free vinorelbine and its active metabolites in relation to liver function in clinical studies of sorafenib/vinorelbine in patients with HCC. Our previous studies have shown that when given in combination with fibroblast growth factor receptor (FGFR) inhibitors, such as infigratinib [21] or the FGFR4 inhibitor FGF401 [22], vinorelbine effectively inhibited tumor growth and improved the OS of HCC PDX models.

FAK plays a central role in integrin-mediated cell adhesion and signaling [23,24,25]. Autophosphorylation of FAK at tyrosine 397 serves as a molecular switch, triggering a cascade of downstream signaling pathways that regulate cell adhesion, spreading, migration, survival, proliferation, cell cycle progression, and angiogenesis [26,27,28]. In various tumors, including HCC, elevated FAK expression is associated with tumor progression and metastasis [29,30,31]. The overexpression of FAK in HCC significantly correlates with an increased risk of extrahepatic metastasis (*p* = 0.027) and a reduced 5-year OS rate (*p* = 0.017). Consequently, the inhibition of FAK is considered a potential therapeutic strategy for patients with HCC [32].

The combination of vinorelbine and cisplatin is extensively utilized in the treatment of non-small-cell lung cancer (NSCLC); however, drug resistance is considered a primary contributor to treatment failures in over 90% of patients with metastatic cancers [33]. The FAK pathway has been implicated in approximately 25% of cases where lung cancer exhibits resistance to vinorelbine, cisplatin, and the combination of cisplatin plus paclitaxel [34]. This pathway plays a crucial role in promoting vinorelbine resistance in lung cancer cells [35]. Notably, it has been reported that IMB5046, vinorelbine, combretastatin A-4-phosphate, and other microtubule-depolymerizing agents (MDAs) induce membrane blebbing in human endothelial cells, leading to vessel disruption and damage through the activation of FAK [36,37].

In the present study, we investigated the potential efficacy of combining sorafenib with vinorelbine as an alternative treatment strategy for HCC patients using 22 HCC PDX models.

## 2. Results

### 2.1. Dose-Dependent Antitumor Activity of Vinorelbine in the HCC13–0109 PDX Model

As shown in Figure 1A,B, vinorelbine exhibited a dose-dependent reduction in the growth of HCC13–0109 PDX model. The tumor burden exhibited reductions of 20%, 59%, and 78% for the vinorelbine dose of 1 mg/kg (equivalent to 0.01 mg/m^2^), 2 mg/kg (equivalent to 0.02 mg/m^2^), and 3 mg/kg (equivalent to 0.03 mg/m^2^), respectively (Figure 1C). No significant loss in body weight (Appendix A) or other clinical signs of toxicity, such as a reduction in food intake, drinking, and ruffed fur, were observed in the mice receiving vinorelbine compared to the vehicle. Consistent with our prior studies [21,22], a dose of 3 mg/kg of vinorelbine administered twice per week demonstrated robust efficacy while maintaining minimal toxicity. Consequently, this dose was selected for subsequent studies.

### 2.2. Enhancement in Antitumor Activity through the Combination of Vinorelbine with Sorafenib

Our next objective was to determine the optimal dose of vinorelbine to combine with sorafenib, aiming for maximal antitumor activity with minimal toxicity. To achieve this, mice bearing HCC13–0109 tumors were treated with a standard dose of 15 mg/kg sorafenib while varying the vinorelbine dosage to 1 mg/kg, 2 mg/kg, and 3 mg/kg. A total of six groups of mice with HCC13–0109 xenografts (*n* = 10/group) received the following treatments: (a) vehicle plus phosphate-buffered saline (PBS), (b) 15 mg/kg sorafenib plus PBS, (c) 3 mg/kg vinorelbine plus vehicle, (d) 15 mg/kg sorafenib plus 1 mg/kg vinorelbine (Sora_15_/Vino_1_), (e) 15 mg/kg sorafenib plus 2 mg/kg vinorelbine (Sora_15_/Vino_2_), and (f) 15 mg/kg sorafenib plus 3 mg/kg vinorelbine (Sora_15_/Vino_3_). The vehicle and sorafenib were administered orally once per day, while PBS and vinorelbine were given intraperitoneally twice per week (once every 3.5 days).

As shown in Figure 1D–F, the Sora_15_/Vino_1_ group did not yield a significant difference in tumor burden compared to those treated with sorafenib or vinorelbine monotherapies. However, the Sora_15_/Vino_3_ group exhibited a remarkable reduction in tumor burden of 15-fold, 4-fold, and 2.5-fold compared to the vehicle, sorafenib, and vinorelbine monotherapies, respectively. These significant reductions in tumor growth and size indicate that the addition of vinorelbine to sorafenib substantially improved the antitumor efficacy of the monotherapies. Similar results were observed when investigating the HCC26–0808B and HCC03–1013 PDX models (Appendix A). The Sora_15_/Vino_3_ group demonstrated superior efficacy (HCC26–0808B; Appendix A) or comparable efficacy (HCC13–0109 and HCC03–1013; Figure 1F and Appendix A) compared to the Sora_15_/Vino_2_ group. No significant loss in body weight (Appendix A) was observed in the drug-treated mice compared to the vehicle-treated mice.

To evaluate the hepatoxicity of the drug treatments, we performed an analysis of liver enzymes in the sera obtained from mice bearing HCC PDX tumors subjected to 16 days of treatment with the vehicle, sorafenib, vinorelbine, and Sora/Vino. As presented in Table 1, the daily administration of sorafenib led to modest increases in alanine aminotransferase (ALT), alkaline phosphatase (ALP), aspartate aminotransferase (AST), and total bilirubin (TBIL). This observation aligns with the safety profiles seen in human studies [5,6], where HCC patients on sorafenib exhibited enzyme elevations. In contrast, vinorelbine induced more substantial elevations in ALT, ALP, AST, and TBIL, indicating mild liver dysfunction. Both sorafenib and vinorelbine monotherapies led to a mild increase in blood urea nitrogen (BUN), approximately 1.25-fold and 1.12-fold, respectively, compared to the vehicle group. Combining sorafenib with vinorelbine resulted in a further elevation in BUN, ALT, ALP, AST, and TBIL, although this increase was not statistically significant compared to vinorelbine monotherapy (Table 1). No significant changes in serum glucose (GLU) and albumin (ALB) levels were observed within the treatment groups compared to those in the vehicle group. These findings collectively suggest that vinorelbine and Sora/Vino induce mild hepatic toxicity. Based on these findings, we selected a dose of 15 mg/kg of sorafenib plus 3 mg/kg of vinorelbine for subsequent combined experiments.

### 2.3. Combination Therapy with Sorafenib and Vinorelbine Demonstrated Effective Antitumor Activity in HCC PDX Models

To further substantiate the antitumor effects of Sora/Vino in HCC, we conducted a comprehensive study involving 22 HCC PDX models with varying levels of sorafenib or vinorelbine sensitivity. Mice bearing specific HCC xenografts were divided into four groups (*n* = 10/group) and received the following treatments: (a) vehicle plus PBS, (b) 15 mg/kg sorafenib plus PBS, (c) 3 mg/kg vinorelbine plus vehicle, and (d) 15 mg/kg sorafenib plus 3 mg/kg vinorelbine (Sora/Vino).

Figure 2 and Appendix A present a summary of the tumor burden observed in 22 HCC PDX models following the drug treatments over a specified period. Appendix A presents the T/C ratio for the sorafenib, vinorelbine, and Sora/Vino groups across the 22 HCC PDX models tested. The threshold to determine the sensitivity of tumors to drug treatment was as follows: tumors with T/C ratios < 0.3 were considered sensitive, those with T/C ratios between 0.3 and 0.42 were considered moderately sensitive, and those with T/C ratios > 0.42 were considered less sensitive (resistant).

For the sorafenib monotherapy, 7/22 (31.8%), 11/22 (50.0%), and 4/22 (18.2%) of the HCC PDX models had T/C ratios < 0.3, between 0.3 and 0.42, and >0.42, respectively, whereas for the vinorelbine monotherapy, 2/22 (9.1%), 6/22 (27.3%), and 14/22 (63.6%) of the HCC PDX models had T/C ratios < 0.3, between 0.3 and 0.42, and >0.42, respectively. For the Sora/Vino combined treatment, 19/22 (86.4%) and 3/22 (13.6%) of the HCC PDX models had T/C ratios < 0.3 and between 0.3 and 0.42, respectively. The Sora/Vino group consistently demonstrated the most superior efficacy across all tested HCC PDX models, exhibiting a significantly decreased tumor burden compared to the vehicle and monotherapy groups.

The effectiveness of the Sora/Vino treatment was particularly notable in HCC01–0207, HCC01–1215, HCC09–0913, and HCC16–1014 PDX models, all of which exhibited relative resistance (with a T/C ratio > 0.42) to sorafenib and vinorelbine monotherapies (Appendix A). A lower *p*-value, when compared to the sorafenib monotherapy, indicated a higher level of statistical significance, reinforcing the potent antitumor activity of combination therapy in suppressing tumor growth in the HCC PDX models (Appendix A).

Appendix A displays photographs of representative HCC PDX models treated with the vehicle, sorafenib, vinorelbine, or Sora/Vino. All experimental mice exhibited a healthy coat, normal food and water intake, regular social interactions and activity levels, and no signs of aggression among cage mates. As illustrated in Appendix A, there was no significant loss in body weight observed in the drug-treated mice compared to the vehicle-treated mice. These findings indicate that the administered dosage resulted in minimal toxicity and side effects.

### 2.4. Combination Therapy Inhibited Angiogenesis and Induced Apoptosis in HCC PDX Models

As sorafenib targets angiogenesis, a crucial process for supplying oxygen and nutrients to tumors and supporting cancer cell survival and proliferation, we conducted immunohistochemistry (IHC) analysis to investigate whether Sora/Vino inhibits tumor angiogenesis, suppresses tumor cell proliferation, and promotes apoptosis.

Tissue sections from HCC13–0109, HCC29–1104, HCC01–0207, HCC15–0114, HCC21–0208, and HCC25–0914 were stained with antibodies against CD31 to assess the degree of tumor angiogenesis, p-histone H3 Ser10 to visualize proliferative cells, and cleaved Poly (ADP-ribose) polymerase (cleaved PARP) to detect apoptotic cells. The representative captured images are presented in Figure 3 and Appendix A, and their quantitative analyses are shown in Appendix A.

Blood vessel density in vinorelbine-treated tumors exhibited significant increases in the HCC25–0914, HCC01–0207, and HCC29–1104 or insignificant changes in the HCC13–0109, HCC15–0114, and HCC21–0208 compared to that of vehicle-treated tumors (Figure 3 and Appendix A). This suggested that the impact of vinorelbine on blood vessels was dependent on the specific tumor line. Structurally, blood vessels in vinorelbine-treated tumors appeared either morphologically slimmer (HCC01–0207, HCC25–0914, and HCC29–1104) or slightly dilated (HCC15–0114 and HCC21–0208) than those in the vehicle-treated tumors (Figure 3 and Appendix A), suggesting that the vinorelbine altered the tumor vasculature in certain HCC PDX models. As expected, both the sorafenib and Sora/Vino treatments significantly reduced the number and size of formed blood vessels compared to the vehicle or vinorelbine monotherapy.

In the HCC01–0207 and HCC13–0109 PDX models, sorafenib demonstrated negligible effect on p-histone H3 Ser10-positive cells compared to the vehicle group (Appendix A; *p* < 0.01). Similarly, in the HCC09–0913, vinorelbine exhibited an insignificant effect on p-histone H3 Ser10-positive cells. Combining sorafenib with vinorelbine did not significantly reduce the positively stained cells of p-histone H3 Ser10 (Appendix A; *p* < 0.01). Notably, in the HCC13–0109, although vinorelbine induced a significant 4.2-fold increase in p-histone H3 Ser10-positive cells, the addition of sorafenib to vinorelbine did not result in a significant decrease in these cells (Appendix A; *p* < 0.01). Conversely, in the HCC16–1014, HCC25–0705A, and HCC29–1104 PDX models, vinorelbine treatment led to a significantly higher number of p-histone H3 Ser10-positive cells compared to the vehicle group. However, the addition of sorafenib to vinorelbine significantly reduced the number of p-histone H3 Ser10-positive cells compared to vinorelbine alone (Appendix A; *p* < 0.01). This observation suggests that the combination therapy had a notable effect in arresting cells in the mitotic phase [38], highlighting its potential in modulating cell cycle dynamics in these specific PDX models.

The impact of sorafenib and vinorelbine monotherapies on inducing apoptosis, as determined by the percentage of cleaved PARP-positive cells, displayed variability across various HCC PDX models. In the HCC16–1014, HCC25–0705A, and HCC29–1104 PDX models, both sorafenib and vinorelbine monotherapies exhibited equal potency in inducing apoptosis, resulting in a two- to three-fold increase in apoptotic cells, as compared to the vehicle group (Appendix A). In contrast, as shown in Appendix A, when compared to the vehicle group in the HCC09–0913 and HCC13–0109 PDX models, sorafenib monotherapy showed an insignificant apoptotic effect, while vinorelbine monotherapy demonstrated a significantly high apoptotic effect. In the HCC01–0207 PDX model, vinorelbine exhibited no significant apoptotic activity, but sorafenib monotherapy displayed a more potent apoptotic effect than the vehicle group.

Nevertheless, the combination of sorafenib and vinorelbine led to a significant increase in the number of cleaved PARP-positive cells and tumor necrosis compared to sorafenib or vinorelbine monotherapies across the 22 HCC PDX models tested (representative data are shown in Figure 3, Appendix A). This suggests a synergistic effect of the combination therapy in enhancing apoptotic responses and tumor necrosis, emphasizing its potential as a comprehensive treatment approach across a diverse range of HCC models.

### 2.5. Combination Therapy Reduced FAK Phosphorylation and Inhibited the FAK Pathway in HCC PDX Models

To gain deeper insights into the mechanism(s) by which Sora/Vino exerted its antitumor activity in HCC PDX models, Western blot analyses were performed on the harvested tumors. Figure 4 illustrates the results of the Western blot analysis conducted in the HCC21–0114 PDX model treated with sorafenib, vinorelbine, or Sora/Vino. Sorafenib treatment led to significant reductions in the levels of survivin, p130Cas, p-p70S6K (Thr421/Ser424), p-4EBP1 (Thr70), and p-S6R (Ser235/236). On the other hand, vinorelbine treatment significantly decreased the levels of Cyclin D1 and p-AKT (Ser473) but increased the levels of Rb, survivin, p-p70S6K (Thr389), p-p70S6K (Thr421/Ser424), and p-4EBP1 (Thr70). Sorafenib and vinorelbine acted synergistically, leading to a reduction in the levels of p-FAK (Tyr397), p-FAK (Ser722), p-FAK (Tyr407), p-Cyclin D1 (Thr286), Cyclin D1, p-Rb (Ser780), E2F1, p130Cas, survivin, Shc, Cdc25C, p-AKT (Ser473), p-p70S6K (Thr421/424), p-4EBP1 (Thr70), p-S6R (Ser235/236), and p-eIF4E (Ser209). However, no significant changes were observed in the levels of FAK, E-cadherin, Cyclin A2, Cdc2, p-Cdc2 (Tyr15), p27, ERK1/2, and p-p70S6K (Thr389).

To further validate our findings, we conducted Western blot analyses on tumors treated with the vehicle, sorafenib, vinorelbine, and Sora/Vino across various HCC PDX models using identical antibodies. The results revealed significant variations in specific protein levels among the drug-treated tumors in different PDX models. For example, in the HCC16–1014 PDX model treated with Sora/Vino, there were notable reductions in the levels of various proteins, including p-FAK, FAK, p-Cyclin D1 (Thr286), p-Cdc2 (Tyr15), p-Cdk2 (Thr14/Tyr15), Cdk2, p-AKT (Ser473), E2F1, p130Cas, Shc, p-p70S6K (Thr389), p-4EBP1 (Thr70), p-S6R (Ser235/236), and p-eIF4E (Ser209) (Appendix A). Notably, the levels of Cdc2 and p-ERK1/2 remained unchanged compared to the vehicle or monotherapies. In another example, as shown in Appendix A, Sora/Vino-treated tumors in the HCC13–0212 PDX model exhibited significant reductions in p-FAK and E2F1 levels, with no significant changes in the levels of FAK, p-cyclin D1 (Thr286), Cyclin D1, p-Rb (Ser780), p130Cas, Paxillin, p-p27 (Ser10), p27, p-AKT (Ser473), p-4EBP1 (Thr70), and p-ERK1/2. Additionally, the Western blot results of drug-treated tumors in the HCC24–0309 and HCC27–1014 PDX models are presented in Appendix A, respectively. Consistently observed across various HCC PDX models, Western blot analyses demonstrated significant reductions in the levels of phosphorylated forms of FAK (p-FAK) at Tyr397, Tyr407, and Tyr925 upon the Sora/Vino treatment. Representative Western blot analyses of HCC19–0913, HCC24–0309, HCC27–1014, HCC13–0109, HCC29–1104, HCC25–0705A, HCC29–0909A, and HCC26–0808B tumors using p-FAK and FAK are shown in Figure 5 and Appendix A. Additionally, an increase in the level of cleaved caspase 3 was observed across various HCC PDX models (Figure 5). Given the reported role of the FAK pathway in promoting vinorelbine resistance in lung cancer cells [35], and the induction of membrane blebbing in human endothelial cells by IMB5046, vinorelbine, and combretastatin A-4-phosphate, leading to the vessel disruption and damage through FAK activation [36,37], we propose that the potent antitumor activity of Sora/Vino likely occurs through its targeting of the FAK signaling pathway.

### 2.6. Knockdown of FAK Inhibited the FAK Pathway and Induced Apoptosis

To validate the proposed hypothesis, HCC13–0109 cells underwent transfection with FAK (to overexpress FAK), shFAK (short hairpin FAK to knockdown FAK), and shLuc (short hairpin Luciferase as non-targeting negative control) constructs. As depicted in Figure 6, the introduction of the FAK construct led to an increase in the expression of total FAK protein, p-Shc (Tyr239/240), and p-c-Myc (Ser62) while simultaneously reducing the basal levels of cleaved PARP. The levels of p-ERK1/2 and p-AKT (Ser473) remained unchanged by the overexpression of FAK. Conversely, the knockdown of FAK resulted in a reduction in the levels of p-FAK (Tyr397), p-FAK (Tyr576/577), total FAK, p-Rb (Ser807/811), p-ERK1/2, p-AKT (Ser473), p-p70S6K (Thr421/Ser424), p-Cdc25C (Ser216), Cdc25C, p-c-Myc (Ser62), p-Shc (Tyr239/240), p-S6R (Ser235/236), and survivin, while increasing the expression of p27, p-p27 (Ser10), and cleaved PARP. These findings underscore the essential role of FAK in HCC cell survival and its involvement in regulating proteins associated with cell cycle progression and apoptosis. Therefore, the inhibition of the FAK pathway by Sora/Vino significantly contributes to its antitumor activity.

### 2.7. Combination Therapy Prolonged the Survival Rate of HCC Orthotopic PDX Models

To assess the impact of Sora/Vino treatment on the OS of HCC orthotopic PDX models, four orthotopic models (HCC13–0212, HCC19–0913, HCC25–0705A, and HCC29–0714B) were utilized. Mice were divided into four groups and subjected to treatment with the vehicle, sorafenib, vinorelbine, or Sora/Vino treatment, as described in Section 4. 

The Kaplan–Meier survival analysis, presented in Figure 7, revealed that all mice treated with the vehicle reached a moribund state on Days 47 (HCC13–0212), 58 (HCC19–0913), 53 (HCC25–0705A), and 63 (HCC29–0714B). Both the sorafenib and vinorelbine monotherapies significantly extended the survival time of HCC orthotopic mice (*p* < 0.01, log-rank test). In the HCC13–0212 and HCC19–0913 models, the vinorelbine-treated groups exhibited longer survival times (77 days and 111 days, respectively) than the sorafenib-treated groups (67 days and 107 days, respectively). Conversely, in the HCC25–0705A and HCC29–0714B models, the sorafenib-treated groups had longer survival times (77 days and 100 days) than the vinorelbine-treated groups (70 days and 80 days, respectively). These findings suggest that the efficacy of the sorafenib and vinorelbine monotherapies in prolonging the survival of HCC orthotopic models was model-dependent. Furthermore, there were no significant differences in survival time between the sorafenib- and vinorelbine-treated mice (*p* < 0.05, log-rank test). The groups treated with Sora/Vino had the longest survival time. Mice in the Sora/Vino group survived until Days 120, 173, 113, and 138 for the HCC13–0212, HCC19–0913, HCC25–0705A, and HCC29–0714B models, respectively (*p* < 0.01, log-rank test). These data demonstrate that Sora/Vino surpasses sorafenib or vinorelbine monotherapies in improving the OS of mice with HCC orthotopic tumors.

## 3. Discussion

The diagnosis of HCC is associated with a grim prognosis and is unresponsive to existing therapeutic modalities [39]. More than 80% of HCCs arise within chronic liver diseases resulting from viral hepatitis, alcohol use, non-alcoholic fatty liver disease (NAFLD), hemochromatosis, obesity, metabolic syndrome, or exposure to genotoxins [40]. Strong evidence indicates that the presence of intratumor heterogeneity is a common feature in HCC, with several stable molecular subtypes found, making clinical management of HCC challenging. Recent studies have suggested that this heterogeneity is partly attributed to the existence of a diversity of hepatic cancer stem cell (CSC) subpopulations. Accumulating evidence has shown that CSCs are involved in tumorigenesis, local recurrence, and the development of therapeutic drug resistance in HCC [41,42,43,44]. While sorafenib has shown some efficacy in improving OS in HCC patients, its effects often prove to be transient. Preclinical and clinical evaluations of sorafenib against cancer-driver pathways indicate that HCC tumors frequently exhibit inherent or acquired resistance. Given the intricate and heterogeneous nature of HCC, comprising a complex mixture of cancer cells, immune cell populations, and stromal cells, durable responses often necessitate combination therapies. Due to its mechanism of action, favorable safety profile, and tolerability, sorafenib has emerged as a promising candidate for synergistic combinations with other anticancer agents possessing complementary mechanisms of action, aiming to improve outcomes for patients with HCC.

We selected vinorelbine to complement sorafenib, as vinorelbine inhibits microtubule polymerization and has demonstrated the ability to enhance the antitumor effects of FGFR inhibitors [21,22] or radiation therapy in HCC models [45], even at lower doses. In this study, we explored the potential synergistic effects of sorafenib and vinorelbine, considering clinically relevant doses, therapeutic indices, and cost-effectiveness. Our approach involves robust preclinical HCC models to assess the efficacy and safety of co-administering sorafenib with vinorelbine (Sora/Vino). Furthermore, we aimed to gain deeper insights into the molecular mechanisms underlying the antitumor effects of this combination. In this report, we present the impact of sorafenib, vinorelbine, and the combined Sora/Vino treatment on tumor growth, tumor angiogenesis, and apoptosis in human HCC PDX models. Further investigation is required to ascertain whether the significant inhibition of the FAK signaling pathway resulted from the combined effect of sorafenib and vinorelbine inducing more cell death, or if the apoptotic impact of vinorelbine, in conjunction with the antitumor and antiangiogenesis activities of sorafenib, led to increased cell death, thereby causing the observed inhibition in the FAK signaling pathway. This intensified inhibition of angiogenesis, coupled with a substantial increase in apoptosis, likely contributes to the potent antitumor activity observed upon combination treatment.

The present study aimed to explore the therapeutic potential and synergistic effects of combining sorafenib and vinorelbine in HCC treatment using a comprehensive set of 22 HCC PDX models. The dosing strategy in our study involved fixing the dose of sorafenib, which targets the main driver pathways, at 15 mg/kg to optimize antitumor cell killing while escalating the dose of vinorelbine, which augments the combination’s effectiveness. To further enhance the tolerability of the Sora/Vino combination, our exploration of various dosing schedules is critical. The use of continuous sorafenib dosing, often optimal for combination strategies, paired with a pulsatile vinorelbine administration schedule (e.g., 1 day on, 3.5 days off) is designed to mitigate toxicity while maximizing antitumor efficacy. This approach was meticulously evaluated within our preclinical setting. Figure 1 and Appendix A illustrate that vinorelbine, in a Sora/Vino combination, is responsible for the observed antitumor effects. Presenting the results from these PDX studies as a percentage of tumor growth inhibition relative to control, along with data on absolute tumor volumes and their temporal changes, enabled us to interpret the data and infer the relative roles of both agents in mediating the observed therapeutic effects. Analysis of liver enzymes in sera obtained from HCC PDX models treated with sorafenib, vinorelbine, and Sora/Vino revealed modest increases in ALT, ALP, AST, TBIL, and BUN (Table 1), suggesting that vinorelbine and Sora/Vino induced mild liver dysfunction and hepatic toxicity in mice. This finding aligns with clinical reports where vinorelbine treatment was associated with serum aminotransferase level elevations in 5% to 10% of patients [46], and high-grade toxicities and poor tolerance were observed in patients with moderate and severe liver dysfunction when administered vinorelbine at doses of 15 mg/m^2^ and ≥7.5 mg/m^2^, respectively [20].

In this study, we present compelling evidence indicating that the addition of vinorelbine to sorafenib consistently yields exceptional therapeutic efficacy in preclinical studies using HCC PDX models. Further clinical studies are required to validate this observation. Notably, the antitumor activity of Sora/Vino significantly outperforms the effects of single agents in most of the HCC models tested. The significantly lower T/C ratio in the Sora/Vino group, compared to the monotherapy or vehicle groups, robustly substantiates the enhanced efficacy of the combination treatment. Furthermore, survival analysis demonstrated a significantly prolonged OS rate in the HCC orthotopic PDX models treated with Sora/Vino, strengthening the evidence for the efficacy of this combination approach over monotherapy or control groups in HCC PDX models. Importantly, mice treated with Sora/Vino displayed a healthy appearance, normal food and water intake, and overall behavior (Appendix A), suggesting that the administered dose and treatment schedule of Sora/Vino resulted in minimal adverse effects on the treated mice. These observations warrant further investigation of Sora/Vino in clinical applications.

Moreover, the combined treatment exhibits superior inhibitory and antiangiogenic effects as well as enhanced induction of apoptosis, compared to monotherapy or control. Significant reductions in microvessel density and blood vessel size indicate the inhibition of tumor angiogenesis, whereas the increased expression of cleaved PARP suggests enhanced induction of apoptosis in tumor cells. Furthermore, necrosis becomes more prominent in sorafenib-treated tumors when vinorelbine is added. These findings suggest that the combination therapy acts through dual mechanisms, resulting in decreased tumor angiogenesis and increased tumor cell apoptosis and necrosis to suppress tumor growth.

The results obtained from the IHC analysis, in which the blood vessels in vinorelbine-treated tumors exhibit either morphological slimness (HCC25–0914, HCC01–0207, and HCC29–1104) or are slightly dilated (HCC15–0114 and HCC21–0208) compared to the blood vessels in vehicle-treated tumors (Figure 3 and Appendix A), indicate that vinorelbine indeed induces alterations in tumor vasculature within specific HCC PDX models. However, the exact mechanisms underlying the capacity of vinorelbine to induce changes in blood vessels remain to be fully elucidated. The elevation in intertumoral blood vessel density observed post vinorelbine treatment is likely attributed to the accumulation of bone marrow-derived cells (BMDCs) recruited from adjacent tissues due to the influence of vinorelbine. Prior research has shown that the accumulation of BMDCs within tumors can stimulate the development of new blood vessels, aiding in vasculature recovery [47,48]. It remains to be determined whether these capillary-like blood vessels induced by vinorelbine treatment are well-perfused, functional blood vessels capable of mitigating tumor hypoxia. Although the pro-vascular effect from recruited BMDCs has been implicated in both tumor protection and disease relapse [48,49], the augmentation of anti-angiogenic activity due to the addition of vinorelbine to sorafenib can potentially be attributed to the downregulation of the FAK signaling pathway, which contributed to the improved tumor response.

The mechanism(s) responsible for the potent antitumor activity and antiangiogenic effects of Sora/Vino are yet to be fully elucidated. Notably, Raf-1 plays a pivotal role in maintaining the survival of endothelial cells during angiogenesis, and sorafenib inhibits critical players, such as Raf isoforms, VEGFRs, and PDGFR-β. Notably, FAK activation follows the inhibition of the RAS/RAF/MEK pathway in several preclinical tumor models [50,51,52], a phenomenon also observed in the analysis of patient tumors [53,54]. In the current study, we did not observe such negative regulation in sorafenib-treated tumors. The effects of vinorelbine on FAK expression and its phosphorylation demonstrated model-dependent outcomes. Vinorelbine upregulated FAK expression and phosphorylation in HCC16–1014 (Appendix A) but inhibited FAK phosphorylation in HCC19–0913, HCC24–0309, HCC27–1014, HCC13–0109, (Figure 5), HCC13–0212, (Appendix A), HCC29–1104, HCC29–0909A, and HCC26–0808B (Appendix A) PDX models. Interestingly, significant changes were not observed in FAK and its phosphorylation in HCC21–0114 (Figure 4) and HCC25–0705A (Appendix A) PDX models following vinorelbine treatment. This observation did not show a correlation with the inhibition of the Raf/ERK pathway. The Sora/Vino combination effectively deactivated FAK by inhibiting the tyrosine phosphorylation of FAK at various critical sites, without affecting the total FAK levels. Figure 8 shows the possible mechanisms responsible for the antitumor and antiangiogenic activity of Sora/Vino in HCC PDX models.

FAK plays a crucial role in upregulating VEGFR2 expression in endothelial cells, thereby promoting angiogenesis in triple-negative breast cancer [55]. Furthermore, the cooperation between FAK and Krüppel-like factor 8 (KLF8) enhances VEGFA expression, contributing to angiogenesis and tumor growth [56]. In animal models with implanted human cancer cells, the pharmacological inhibition of FAK prevents angiogenesis and suppresses tumor progression [57,58,59,60,61]. These findings collectively underscore FAK’s pivotal role in angiogenesis for tumor growth [62]. Previous studies have demonstrated that FAK phosphorylation at tyrosine 397 is crucial for many established functions of FAK, including the promotion of cell spreading, migration, cell cycle progression, and cell survival [26,27,28]. Elevated FAK expression has been associated with tumor progression and metastasis in HCC [29,30,31], along with a reduced 5-year OS rate (*p* = 0.017). Our current study shows that combination therapy potently inhibits angiogenesis, certain cell cycle regulators, and the FAK signaling pathway, which plays a significant role in cell survival, proliferation, angiogenesis, and apoptosis. The knockdown of FAK by shRNA provides valuable insights into the critical roles of FAK in HCC and supports the hypothesis that Sora/Vino inhibits angiogenesis and induces apoptosis in various HCC models tested by targeting the FAK pathway.

Since prolonged dosing of sorafenib is typically required for targeted therapies, careful consideration of both delayed and cumulative toxicities is essential when determining suitable dosages for subsequent developmental stages. Our current study effectively demonstrates that combining vinorelbine with sorafenib does not compromise efficacy or significantly increase the toxicity associated with the standard targeted agent, sorafenib. This observation is consistent with a previous study (as reviewed in [63]). By using clinically relevant HCC tumor models and treatment scenarios, we provide compelling evidence of an enhanced tumor response without additional toxicity when a low dose of vinorelbine is added to standard sorafenib. This study holds particular significance, especially when considering the substantial monotherapy activity that sorafenib has already demonstrated in HCC [6,7], alongside the efficacy of vinorelbine in other cancer types [14,15,18,19]. Moreover, this approach is cost-effective, utilizing two established and simple-to-deliver treatment modalities, which makes it accessible and affordable for patients. Further investigation is needed to better understand the interactions between sorafenib and vinorelbine in tumor response and the mechanisms underlying their antitumor and antiangiogenic activities. In summary, this study provides a strong rationale for future phase I/II clinical trials of sorafenib combined with a metronomic dose of vinorelbine, aimed at improving the efficacy of frontline therapy for HCC patients who have previously experienced disease progression while undergoing sorafenib treatment.

## 4. Materials and Methods

The reagents, HCC cell isolation and cultures, Western blot analysis, orthotopic models, immunohistochemistry (IHC), slide imaging and quantification, and statistical analyses were prepared or performed as previously described [21,64,65,66].

### 4.1. Reagents

Sorafenib (Nevaxar^®^) was suspended in a vehicle solution containing 5% glucose, 15% PEG300, and 35% Captisol^®^. Vinorelbine (Navelbine^®^) (10 mg/mL) was obtained from Pierre Fabre Medicament (Boulogne, France) and was dissolved in PBS to a final concentration of 0.375 mg/mL before use.

All primary antibodies used in the Western blot analyses are listed in Appendix A.

### 4.2. Cell Culture

HCC13–0109 cells were isolated from HCC13–0109 tumors and cultured as monolayer cultures in high-glucose Dulbecco’s modified Eagle’s medium (DMEM) supplemented with 10% fetal bovine serum (FBS) and 1% penicillin–streptomycin at 37 °C with 5% CO_2_, as previously described [64].

### 4.3. HCC Patient-Derived Xenograft (PDX) Models

This study received ethics board approval from the SingHealth Centralised Institutional Review Board (ethics code: CIRB #2006/435/B; approval date: 2 October 2018). All the animals were maintained in accordance with the guidelines outlined in the Guide for the Care and Use of Laboratory Animals, published by the National Institutes of Health, USA [67].

HCC PDX models were generated in male C.B-17 severe combined immunodeficiency (SCID) mice aged from 9 to 10 weeks old and with a body weight of 23–25 g (InVivos Pte. Ltd., Singapore), as previously described [65]. Briefly, under sterile conditions, the HCC tumors were minced into fine fragments that would pass through an 18-gauge needle and then mixed at a ratio of 1:1 (*v*/*v*) with Matrigel^®^ (Corning Inc., Corning, NY, USA) to result in a total volume of 150 μL per injection. The tissue mixture was subcutaneously injected in both flanks of each mouse. The growth of the xenograft tumors was monitored at least twice weekly until the tumor sizes reached approximately 170–200 mm^3^.

Mice were housed in negative-pressure isolators set at 23 °C and 43% humidity, with 12 h light/dark cycles, and were provided with sterilized food and water ad libitum. All studies were performed in accordance with IACUC-approved procedures.

### 4.4. Drug Treatment and Efficacy of Sora/Vino in 22 Ectopic HCC PDX Models

For the dose–response experiment, groups of five mice bearing HCC13–0109 tumors were treated intraperitoneally twice per week (once every 3.5 days) with the vehicle (PBS) or vinorelbine at doses of 1 mg/kg (equivalent to 0.01 mg/m^2^), 2 mg/kg (equivalent to 0.02 mg/m^2^), or 3 mg/kg (equivalent to 0.03 mg/m^2^). For the combination therapy, mice bearing the indicated HCC PDX models (22 models in total) were randomized into four treatment groups (*n* = 10) and treated as follows: (a) vehicle plus PBS; (b) 15 mg/kg of sorafenib plus PBS; (c) 1 mg/kg, 2 mg/kg, or 3 mg/kg of vinorelbine plus the vehicle; or (d) 15 mg/kg of sorafenib plus 1 mg/kg, 2 mg/kg, or 3 mg/kg vinorelbine for the indicated time. The vehicle and sorafenib were administered orally daily, while the PBS and vinorelbine were administered intraperitoneally twice per week (once every 3.5 days).

All the treatment started when the tumor sizes reached approximately 170–200 mm^3^. The tumor growth and signs of illness were monitored and recorded, as described in previous studies [21,22,45,64,66]. At the end of the experiment, the mice were sacrificed, and the tumors were resected, weighed, and recorded. The harvested tumors were divided into two parts: one part was snap-frozen in liquid nitrogen for molecular analyses, and the other part was fixed in 10% formalin and processed for IHC.

To evaluate the efficacies of the sorafenib, vinorelbine, and Sora/Vino treatments in the HCC PDX models, the T/C ratio was calculated by dividing the median weight of the drug-treated tumors (T) by that of the vehicle-treated tumors (C) at the end of the treatment. In this study, tumors with T/C ratios < 0.3 were considered sensitive, those with T/C ratios between 0.3 and 0.42 were considered moderately sensitive, and those with T/C ratios > 0.42 were considered less sensitive (resistant), in accordance with the criteria established by the Cancer Therapy Evaluation Program (CTEP) of the Investigational Drug Branch (IDB) at the National Cancer Institute [68].

### 4.5. Serum Analysis

Sera were derived from the mice treated with the vehicle, sorafenib, vinorelbine, and Sora/Vino for 16 days. The sera were collected to determine the levels of total bilirubin (TBIL), alkaline phosphatase (ALP), alanine aminotransferase (ALT), aspartate aminotransferase (AST), albumin (ALB), creatinine (Cre), glucose (GLU), and blood urea nitrogen (BUN) using the VETSCAN^®^ Preventive Care Profile Plus (Abaxis Inc., Union City, CA, USA) according to the manufacturer’s instructions.

### 4.6. Efficacies of Sorafenib, Vinorelbine, and Sora/Vino in HCC Orthotopic PDX Models

The HCC13–0212, HCC19–0913, HCC25–0705A, and HCC29–0714B orthotopic models were generated as previously described [66]. Briefly, SCID mice were anesthetized with a ketamine/diazepam solution (50 mg/kg of ketamine hydrochloride; Rotexmedica, Trittau, Germany; and 5 mg/kg of diazepam (Alantic), I.M.). Baytril^®^ at 5 mg/kg was given intramuscularly. Under sterile conditions, a small upper midline laparotomy was performed to exteriorize the left lobe of the liver. Approximately 5 × 10^6^ tumor cells (in 30 μL of medium-Matrigel^®^ mixture) were implanted in the lobe of the liver, using a 27-gauge needle. The incision was closed using a running suture of 5-0 silk. Upon tumor establishment, mice (*n* = 10/group) were treated with the vehicle, sorafenib, vinorelbine, or Sora/Vino following the treatment conditions described above. The treatments commenced when the tumor sizes reached approximately 100–150 mm^3^. The tumor growth, body weight, ascites formation, and OS were monitored and recorded daily. The mice were euthanized upon reaching a moribund state, determined by meeting any of the following criteria: weight loss exceeding 10%, abdominal distension, abnormal posture and breathing, ruffled fur, inability to move, loss of appetite (including diminished eating, drinking, and urination), lack of interaction between mice, and a considerably smaller maximum tumor size, with priority given to the assessment of overall health status.

### 4.7. Immunohistochemistry (IHC)

IHC was performed according to a previously described protocol [64]. The slides were stained with antibodies against CD31 (Cell Signaling Technology, Beverly, MA, USA, #77699), p-histone H3 Ser10 (Cell Signaling Technology, Beverly, MA, USA, #9701), and cleaved PARP (Cell Signaling Technology, Beverly, MA, USA, #5625) to assess the microvessel density, cell proliferation, and apoptosis, respectively. At least 10 fields were randomly captured at a magnification of 100× on each IHC-stained slide using an Olympus BX60 microscope (Olympus, Tokyo, Japan). To quantify the mean of the microvessel density, the p-histone H3 Ser10, and the cleaved PARP cells, all the positively stained cells in the captured images were counted and expressed as a percentage value compared with the total number of cells in that region.

### 4.8. Western Blot Analysis and Quantification Analysis

To determine the changes in the protein expressions between the vehicle- and drug-treated tumors, tumors (*n* = 10/group) from each treatment group were harvested, pooled, and snap frozen. Two pooled tumors from each treatment group were homogenized in the lysis buffer containing 50 mM Tris-HCl (pH 7.4), 150 mM NaCl, 0.5% NP-40, 1 mM EDTA, and 25 mM NaF supplemented with protease inhibitors and 10 mM Na_3_VO_4_. Approximately 80 μg of protein per sample was resolved using sodium dodecyl sulphate–polyacrylamide gel electrophoresis (SDS–PAGE) and transferred to a nitrocellulose membrane, as described in [21]. The blots were incubated with the indicated primary antibodies, followed by horseradish peroxidase-conjugated secondary antibodies. The blots were then visualized with WesternBright ECL HRP substrate (Advansta, Inc., San Jose, CA, USA) and exposed to autoradiography film (Agfa Healthcare, Mortsel, Belgium). The developed films were scanned using a GS-900 Calibrated Densitometer.

For the quantification analysis, the total density of each corresponding protein band was quantified using Image Lab^TM^ software (Version 6.1; BioRad, Hercules, CA, USA), normalized to tubulin (loading control), and expressed as a fold change relative to the vehicle group. A value greater (or lesser) than 1 indicated that the expression level of the protein of interest was greater (or lower) than that in the control group.

### 4.9. Gene Overexpression or shRNA Knockdown by Transfection

The HCC13–0109 cells were transfected with FAK (to overexpress the FAK), shFAK (to knockdown the FAK), and shLuc (non-targeting negative control) constructs, respectively, using Lipofectamine 2000 (Invitrogen, Carlsbad, CA, USA) following the manufacturer’s instructions. In brief, the HCC13–0109 cells were trypsinized, counted, and seeded at a density of 1 × 10^6^ cells per 100 mm dish. On the following day, Lipofectamine 2000 was diluted in Opti-MEM (Invitrogen), mixed with the constructs, and incubated at room temperature for 20 min. The mixture was then added to the cells and incubated at 37 °C with 5% CO_2_ for 2 days. Subsequently, the HCC13–0109 cells were harvested, lysed in the lysis buffer, and subjected to Western blot analysis, as described above.

### 4.10. Statistical Analysis

The differences between the tumor volumes and tumor weights at sacrifice, the means of the p-histone H3 Ser10-, cleaved PARP-, and CD31-positive cells, the expression levels of proteins were compared. The Student’s *t*-test was used for comparisons between two groups. One-way analysis of variance (ANOVA) followed by the Tukey–Kramer post hoc test was used when comparing more than two groups. The error bars are given based on the calculated SD values. For the survival analysis, the log-rank test was used. A *p*-value of <0.05 was considered to be statistically significant.

## Figures and Tables

**Figure 1 ijms-25-01563-f001:**
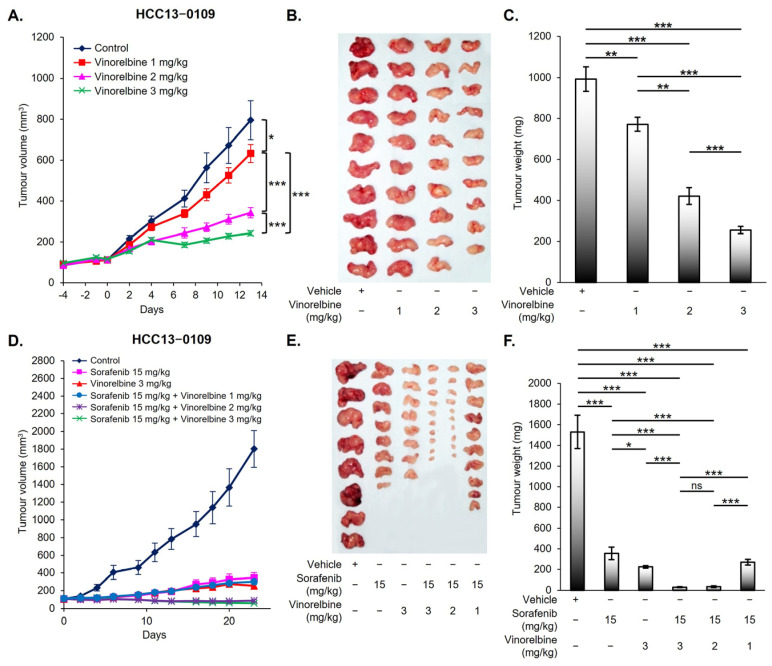
Dose-dependent effects of vinorelbine and Sora/Vino in the HCC13–0109 PDX model. Mice bearing HCC13–0109 xenograft were intraperitoneally treated with the vehicle or vinorelbine at doses of 1 mg/kg, 2 mg/kg, and 3 mg/kg twice per week (every 3.5 days). (**A**) The mean tumor volumes ± standard errors (SEs); (**B**) representative vinorelbine-treated tumors harvested on Day 13; and (**C**) the mean corresponding tumor weights ± SEs at sacrifice are shown. For the combination therapy, HCC13–0109 xenograft mice were treated with the vehicle plus phosphate buffered saline (PBS), sorafenib, vinorelbine, or Sora/Vino as indicated. (**D**) The mean tumor volumes ± SEs; (**E**) representative drug-treated tumors harvested on Day 23; and (**F**) the mean corresponding tumor weights ± SEs are shown. Significant differences were assessed using one-way analysis of variance (ANOVA) followed by Tukey’s test (* *p* < 0.05; ** *p* < 0.01; *** *p* < 0.001; ns, no significance).

**Figure 2 ijms-25-01563-f002:**
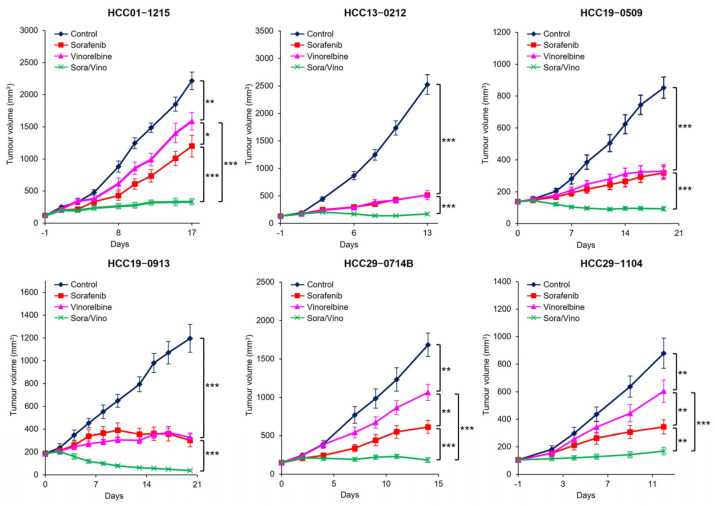
The antitumor effect is enhanced by the combination of vinorelbine with sorafenib. Mice bearing HCC tumors were treated with the vehicle, sorafenib, vinorelbine, or Sora/Vino over a specific period. The mean tumor volumes ± SEs are plotted (* *p* < 0.05; ** *p* < 0.01; *** *p* < 0.001; ANOVA followed by Tukey’s test). The Sora/Vino exhibited a significant reduction in tumor growth compared to the vehicle or monotherapies.

**Figure 3 ijms-25-01563-f003:**
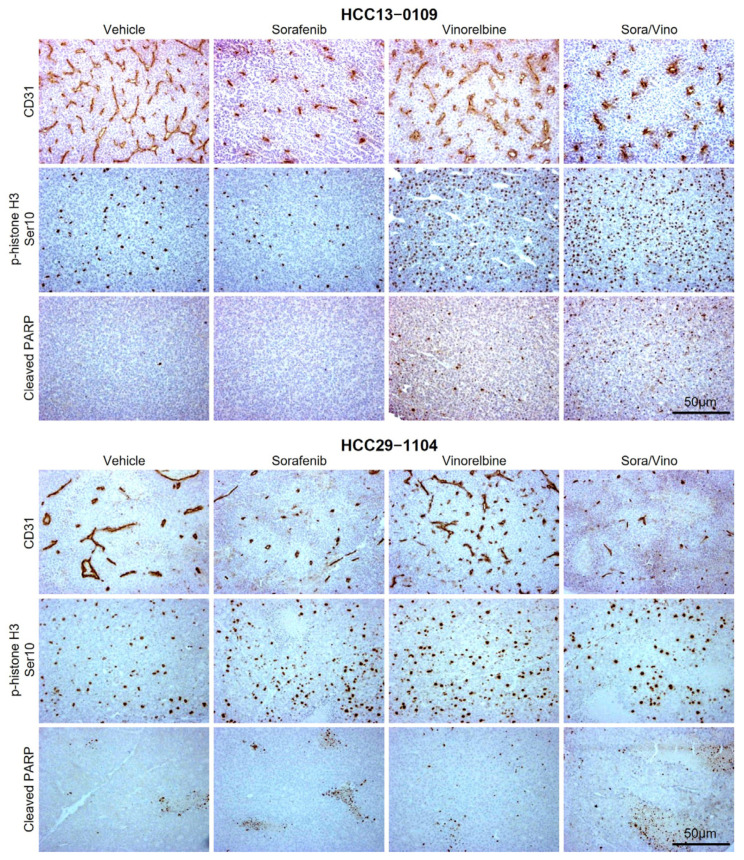
Sora/Vino treatment inhibited angiogenesis and induced apoptosis in HCC PDX models. Mice bearing HCC13–0109 and HCC29–1104 tumors were treated with the vehicle, sorafenib, vinorelbine, or Sora/Vino. Harvested tumors were processed for IHC, as described in Section 4. Representative images of tumor sections from vehicle- and drug-treated mice stained for CD31 (blood vessels), p-histone H3 Ser10, and cleaved PARP are shown. Sorafenib treatment reduced the number and size of blood vessels, vinorelbine treatment led to a significant increase in p-histone H3 Ser10-positive cells, and Sora/Vino treatment resulted in a significant increase in cleaved PARP-positive cells. Tumor necrosis was also observed in the stained sections of the HCC PDX models.

**Figure 4 ijms-25-01563-f004:**
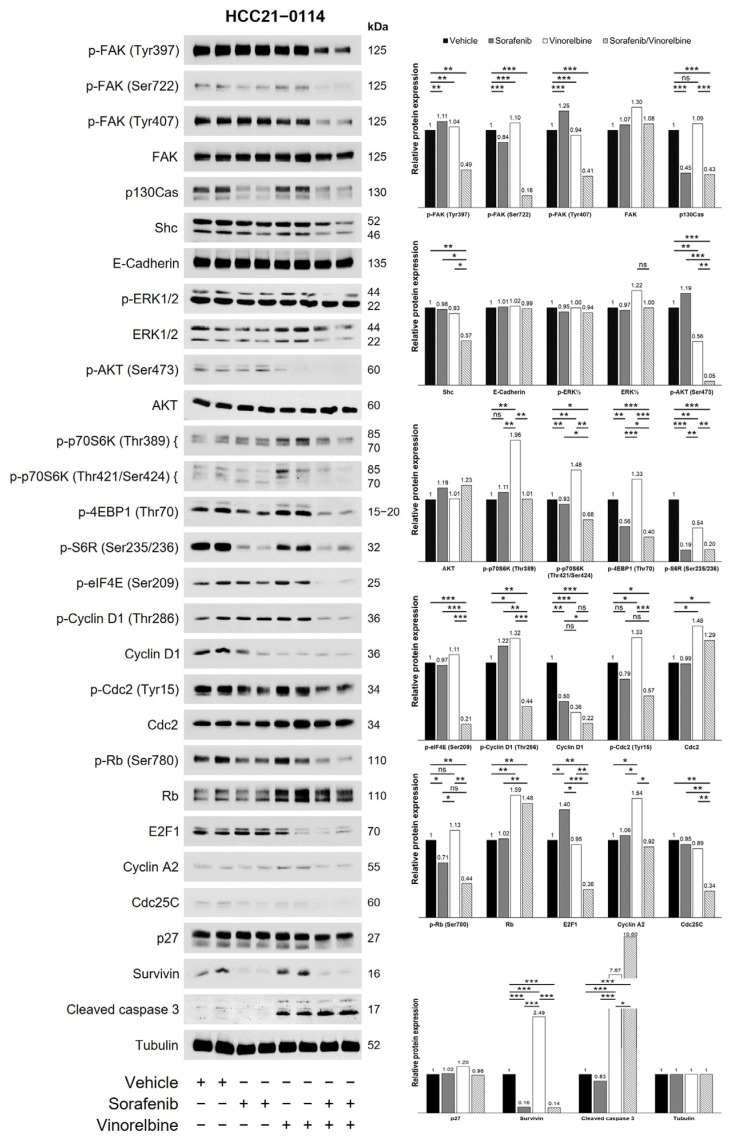
Sora/Vino acted synergistically in the treatment of HCC PDX models. Mice bearing HCC21–0114 tumors were treated with the vehicle, sorafenib, vinorelbine, or Sora/Vino, as described in Section 4. Tumors (*n* = 10/group) from each treatment group were harvested, pooled and snap frozen. Two pooled tumors from each treatment group were subjected to Western blot analysis with the indicated antibodies. The total density of each corresponding protein band was quantified, normalized to tubulin (loading control), and expressed as a fold change relative to the vehicle group. A value greater (or lesser) than 1 indicated that the expression level of the protein of interest was greater (or lower) than that in the control group Statistical analysis was performed using ANOVA followed by Tukey’s test (* *p* < 0.05; ** *p* < 0.01; *** *p* < 0.001; ns, no significance). Western blot analysis demonstrated that HCC21–0114 tumors treated with Sora/Vino exhibited a synergistic effect in reducing the protein levels of p-FAK (Tyr397), p-FAK (Ser722), p-FAK (Tyr407), p-Cyclin D1 (Thr286), Cyclin D1, p-Rb (Ser780), p-AKT (Ser473), survivin, E2F1, p130Cas, Shc, Cdc25C, p-p70S6K (Thr421/424), p-4EBP1 (Thr70), p-S6R (Ser235/236), and p-eIF4E (Ser209) compared to the vehicle or monotherapies groups.

**Figure 5 ijms-25-01563-f005:**
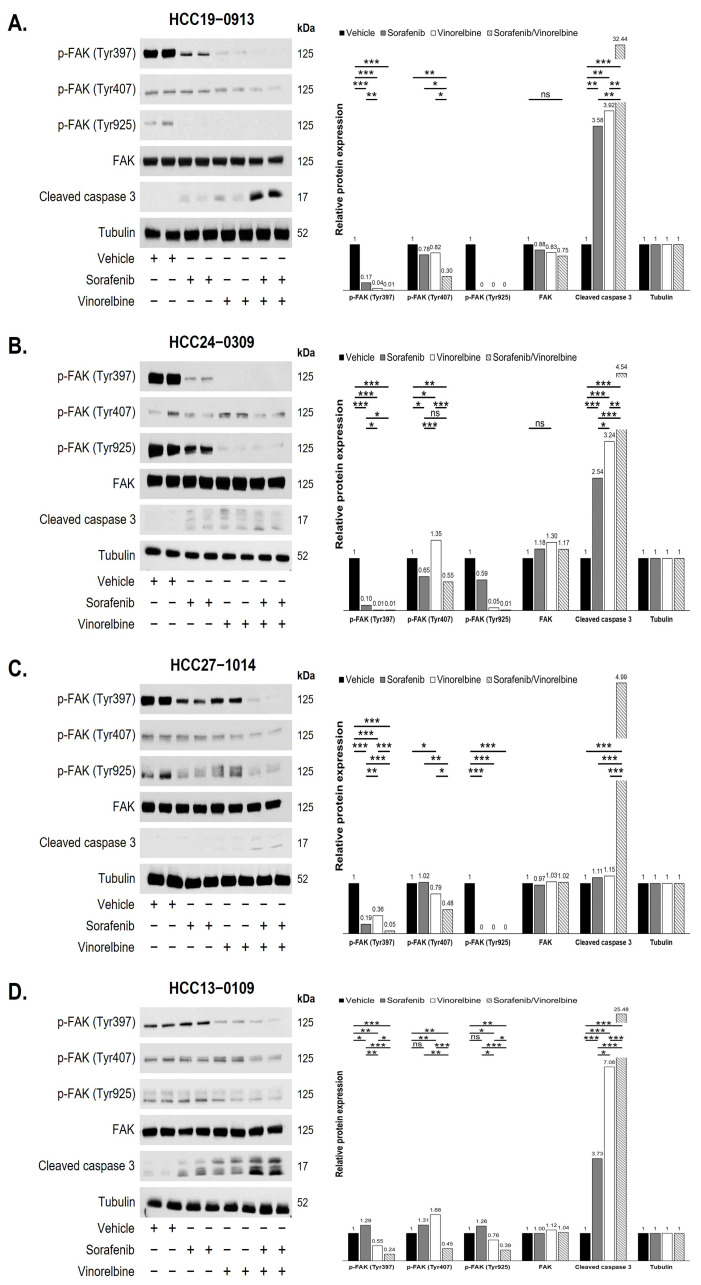
Sora/Vino consistently reduced the FAK phosphorylation in HCC PDX models. Mice bearing the indicated tumors were treated with the vehicle, sorafenib, vinorelbine, or Sora/Vino, as described in Section 4. Tumors (*n* = 10/group) from each treatment group were harvested, pooled and snap frozen. Two pooled tumors from each treatment group were subjected to Western blot analysis with the indicated antibodies. The total density of each corresponding protein band was quantified, normalized to tubulin (loading control), and expressed as a fold change relative to the vehicle group. A value greater (or lesser) than 1 indicated that the expression level of the protein of interest was greater (or lower) than that in the control group Statistical analysis was performed using ANOVA followed by Tukey’s test (* *p* < 0.05; ** *p* < 0.01; *** *p* < 0.001; ns, no significance). Treatment with Sora/Vino in (**A**) HCC19–0913, (**B**) HCC24–0309, (**C**) HCC27–1014, and (**D**) HCC13–0109 significantly decreased the levels of phosphorylated FAK at Tyr397, Tyr407, and Tyr925 compared to the vehicle or monotherapies groups.

**Figure 6 ijms-25-01563-f006:**
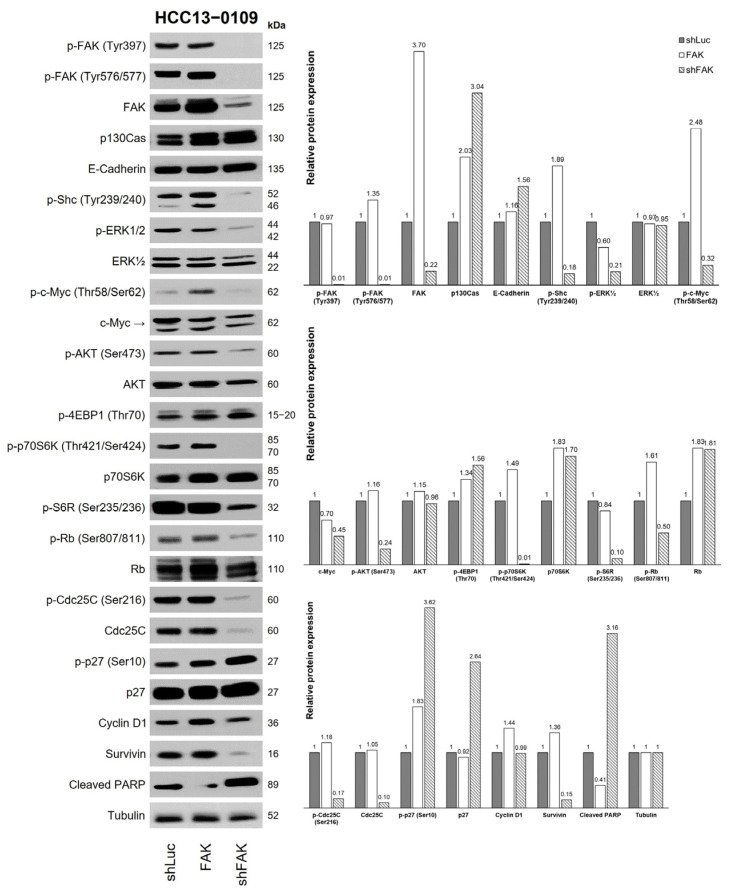
Knockdown of FAK inhibited the FAK pathway and induced apoptosis. HCC13–0109 cells were transfected with FAK, shFAK, and shLuc constructs, as described in Section 4. Transfected cells were collected, lysed in lysis buffer, and subjected to Western blot analysis with the indicated antibodies, as described in Section 4. The total density of each corresponding protein band was quantified, normalized to tubulin (loading control), and expressed as a fold change relative to the vehicle group. A value greater (or lesser) than 1 indicated that the expression level of the protein of interest was greater (or lower) than that in the control group. Statistical analysis was performed using ANOVA followed by Tukey’s test. Knockdown of FAK in HCC13–0109 cells resulted in the reduction levels of p-FAK (Tyr397), p-FAK (Tyr576/577), FAK, p-Rb (Ser807/811), p-ERK1/2, p-AKT (Ser473), p-p70S6K (Thr421/Ser424), p-Cdc25C (Ser216), Cdc25C, p-c-Myc (Ser62), p-Shc (Tyr239/240), p-S6R (Ser235/236), and survivin while increasing the expression of p-p27 (Ser10), p27, and cleaved PARP. These findings suggest the essential role of FAK in HCC cell survival, cell cycle progression, metastasis, and apoptosis.

**Figure 7 ijms-25-01563-f007:**
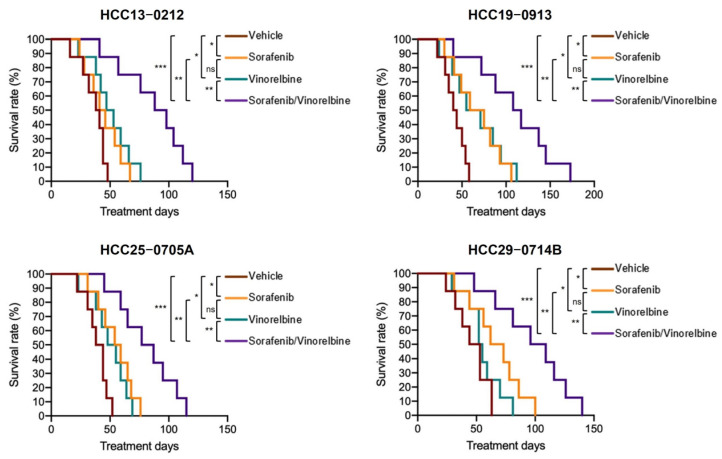
Efficacy of sorafenib, vinorelbine, and Sora/Vino in the HCC orthotopic PDX models. Orthotopic models of HCC13–0212, HCC19–0913, HCC25–0705A, and HCC29–0714B were prepared and treated with the vehicle, sorafenib, vinorelbine, or Sora/Vino, as described in Section 4. Mice were sacrificed when they reached a moribund state. Kaplan–Meier survival analysis showed that Sora/Vino treatment significantly improved the OS of mice with HCC orthotopic tumors (* *p* < 0.05; ** *p* < 0.01; *** *p* < 0.001; log-rank test).

**Figure 8 ijms-25-01563-f008:**
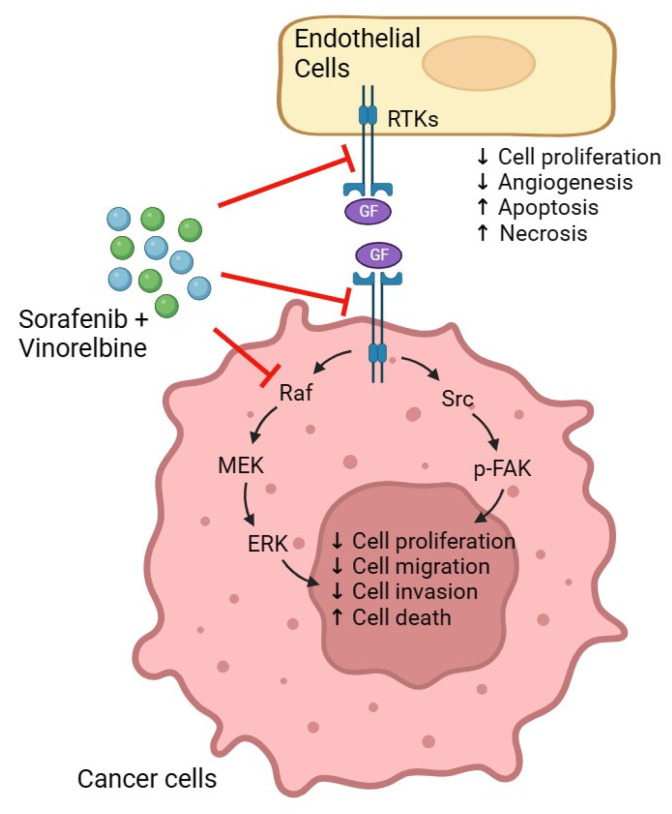
Effects of Sora/Vino on the FAK pathway. Sorafenib acts to inhibit receptor tyrosine kinases (RTKs), such as VEGFR-2, VEGFR-3, PDGFR-β, Flt-3, Ret, and c-kit. When vinorelbine is added to sorafenib, it augments the inhibition of FAK activation in endothelial cells and cancer cells, resulting in the suppression of FAK phosphorylation, leading to a decrease in cell proliferation, migration, invasion, and angiogenesis while promoting apoptosis and tumor necrosis.

**Table 1 ijms-25-01563-t001:** Effects of sorafenib, vinorelbine, and Sora/Vino on liver- and kidney-injury-related parameters. Sera derived from mice bearing HCC13-0109 treated with the vehicle, sorafenib, vinorelbine, and Sora/Vino for 16 days were analyzed using VETSCAN^®^ Preventive Care Profile Plus (Abaxis Inc., Union City, CA, USA) according to the manufacturer’s instructions. The levels of total bilirubin (TBIL), alkaline phosphatase (ALP), alanine aminotransferase (ALT), aspartate aminotransferase (AST), and albumin (ALB) served as markers of overall liver function. The levels of creatinine (Cre), glucose (GLU), and blood urea nitrogen (BUN) served as indicators of kidney function.

Serum Marker	Unit	Vehicle	Sorafenib 15 mg/kg	Vinorelbine 3 mg/kg	Sora/Vino
BUN	(mg/dL)	14.1	17.6	15.8	17.1
CRE	(mg/dL)	0.47	0.58	0.52	0.55
ALT	(U/L)	36.9	55.3	58.7	64.8
ALP	(U/L)	52.7	61.6	89.6	94.3
AST	(U/L)	206	277.5	285.5	312.4
TBIL	(mg/dL)	0.34	0.39	0.42	0.43
GLU	(mg/dL)	157.8	162.3	152.4	171
ALB	(g/dL)	4.25	3.58	3.38	3.49

## Data Availability

The datasets used and analyzed in the current study are available within the manuscript and its Appendix A.

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
