# Peer review of "Vinorelbine Improves the Efficacy of Sorafenib against Hepatocellular Carcinoma: A Promising Therapeutic Approach"

_ijms, 2024, doi:10.3390/ijms25031563_

Round 1

Reviewer 1 Report (New Reviewer)

Comments and Suggestions for Authors

Dear authors, I have carefully reviewed your manuscript entitled “Vinorelbine improves the efficacy of sorafenib against hepatocellular carcinoma: A promising therapeutic approach”, authored by Wai Har Ng, Khee Chee Soo and Hung Huynh.

Although the manuscript is of good quality, more work needs to be done before it is suitable for publication. I have provided detailed point-by-point comments below for the authors' consideration and response.

Line 54, references 16, 17 and line 54 references 18 and 19: these references support the conclusion that that vinorelbine administration results in an objective responses of prolonged duration and minimal toxicity. However, these studies do not take into consideration that the liver metabolize vinorelbine and in patients with liver dysfunction vinorelbine is poorly tolerated (Gong et al 10.1634/theoncologist.2018-0336). Please modify your introduction and include in the discussion.

Line 87 the authors provide the dosage for vinorelbine in mg/Kg (i.e., 1 mg/kg, 2 mg/kg, and 3 mg/kg). However, the dosage of vinorelbine in patients is administered in mg/m2. Please translate the dosage from mg/Kg to mg/m2 so that a direct comparison with the dosage in humans can be evaluated.

Line 127, please include in the supplementary material the table for the animals body weight. Did the author measure liver functionality/hepatotoxicity by measuring liver parameters (e.g., bilirubin, albumin, urea, ALT, GLDH)? Please include.

Line 400, I disagree with the conclusion that "Sora/Vino further suppressed the FAK pathway, leading to an increase in cleaved caspase 3". Vinorelbine is an inhibitor of microtubule polymerisation and this is its MOA. My interpretation is that vinorelbine mediated blockade of microtubule polymerisation in proliferating cancer cells is the main factor driving cell death (as shown by increased caspase 3 cleavage) and the removal of these cells, in combination with sorafenib activity, is what drives the observed reduction in FAK signalling.

Line 420 the statement “In our validation study, we present compelling evidence that the addition of vinorelbine to sorafenib consistently yields exceptional therapeutic efficacy in treating HCC” is misleading. The authors have “only” shown that vinorelbine/sorafenib is effective in reducing tumor development in a PDX model. To support their statement, the author should show that administration of vinorelbine/sorafenib could achieve the same effect in animal models of liver cancer. Please rephrase.

The meaning of some of the abbreviation (e.g., PARP) is not included in the manuscript. List of abbreviations is missing, please include.

Author Response

Reviewer 2 Report (New Reviewer)

Comments and Suggestions for Authors

Main question is answered - vinorelbine improves efficacy in animal model in early phase clinical trials however, it needs to be validated with more clinical trials.

The concept of adding vinorelbine with sorafenib is not very common and should be investigated like this paper tried.

Lot of research needs to be done in the field of HCC medical oncology. This paper is another research in that direction, though not very novel.

HCC patient derived Xenograft models were used for the study, we need to do the study in humans now othewise, no specific changes for methodology in present study is recommended.

Conclusion has answered all the main questions posed, the authors wanted to find. Statistics is well documented.

Can you add role of vinorelbine with other therapies like radiotherapy.

Yeoh KW, Prawira A, Saad MZB, Lee KM, Lee EMH, Low GK, Mohd Nasir MHB, Phua JH, Chow WWL, Lim IJH, Omar YB, Ho RZW, Le TBU, Vu TC, Soo KC, Huynh H. Vinorelbine Augments Radiotherapy in Hepatocellular Carcinoma. Cancers (Basel). 2020 Apr 3;12(4):872.

Round 2

Reviewer 1 Report (New Reviewer)

Comments and Suggestions for Authors

The authors have taken my suggestions into careful consideration and have significantly improved the manuscript. I am pleased to report that the revised version is now suitable for publication in its current form.

This manuscript is a resubmission of an earlier submission. The following is a list of the peer review reports and author responses from that submission.

Round 1

Reviewer 1 Report

Comments and Suggestions for Authors

The article “Vinorelbine improves the efficacy of sorafenib against hepatocellular carcinoma: A promising therapeutic approach” is a very interesting study. Here are some of my concerns through which the manuscript can be improved:

Major concerns

1.     Explain in detail the methods used to generate orthotopic HCC xenograft model in C.B-17 SCID mice in the materials and methods section. Also provide details of the number of HCC cells injected per mice.

2.     As per Fig 1b, and the number of mice used per group i.e (n=10), vinorelbine only treatment mice survived 100 percent whereas in Figure 1E, there is around 8 tumor images per group, as indicated (n=10) did the mice died during drug treatment. If yes, than sorafenib toxicity percentage would be 20%, but the authors have mentioned in the result section that there was no noticeable clinical signs of toxicity.

3.     Where are the tumor images for the HCC patient derived xenograft orthotopic model for the HCC13-0212, HCC19-0913, HCC25-0705A, and HCC29-0714B. My question is, did the authors perform xenograft from all the 41 patient derived HCC. The datas shown in the figures are from different with every figure making it a lot of confusion and unable to properly correlated the datas with the 41 different HCC PDX. I understand the efficacy of drug varied among different PDX models, but there are 41 PDX models here and there is no doubt about the therapeutic efficacy of the sorafenib and vinorelbine but datas in Figure provided are very inconsistent and focused on just a few PDX HCC’s for figures 2-7. Could it be better if the authors reduce the PDX model numbers according to the datas they have provided in the figures as it will create confusion among readers. And providing all 41 PDX datas would be an enormous data for one manuscript.

Minor concerns

1.     In Figure 1b and C Legends are interchanged. Make the necessary correction. Figure 1E legends is incorrectly interpreted as well. Should it be tumor images.

2.     Line 74,85,86, 87, 93,117, 118, 127, 130,141,144,148,149,151,158,161,162,167,169, 174, 176, 178,180, 188,203,204  449, 518 change tumour to tumor. Make the changes throughout the manuscript.

Comments on the Quality of English Language

Minor English editing required

Reviewer 2 Report

Comments and Suggestions for Authors

Overall a very interesting article requiring only a few corrections.

1. In introduction section, the part talking about treatment options should be rephrased, since atezolizumab/bevacizumab is first line treatment for all patients except cirrhotics with  decompensation (relevant references: Singal AG, Hepatology 2023; Vogel A, Ann Oncol 2021)

2. In Discussion section, a small paragraph talking about the diversity of HCC should be added (ref: Desert R, WJG 2018; Androutsakos T, IJMS 2022; Zheng H, Hepatology 2018)

3. A figure with FAK pathway would be helpful to most readers
